# Contrasting long-term temperature trends reveal minor changes in projected potential evapotranspiration in the US Midwest

Bruno Basso [1,2,3,4✉], Rafael A. Martinez-Feria[1,3], Lydia Rill [1] & Joe T. Ritchie[1,4]

Warming generally leads to increased evaporative demand, altering the amount of water needed for growing crops. For the Midwest, some studies have suggested that reaching yield targets by 2050 will not be possible without additional precipitation or large expansion of irrigation. Here, we show that this claim is not supported by the historical summer climate trends, which indicate that the warming of daily average temperatures is largely driven by increases in minimum temperatures, while maximum temperatures have decreased. This has translated into a net decrease in vapor pressure deficit (VPD) and potential evapotranspiration (PET). With the increasing rainfall, this suggests that crop water deficits have likely become less frequent in the region despite the warming climate. By projecting these trends into 2050 and ancillary use of a crop model, we estimate minor changes in PET that would have minimal effects on corn yields (<6%) under persistence of these trends.

[1] Department of Earth and Environmental Sciences, Michigan State University, East Lansing, MI, USA. [2] W.K. Kellogg Biological Station, Michigan State University, Hickory Corners, MI, USA. [3] These authors contributed equally: Bruno Basso, Rafael A. Martinez-Feria. [4] These authors jointly supervised this work: Bruno Basso, Joe T. Ritchie. ✉email: basso@msu.edu

Global near-surface air temperatures have already increased by >1 °C above preindustrial levels[1] and are expected to surpass 1.5 °C of warming by the mid-2050s[2]. Since warmer air can hold more water vapor before it condenses, the increasing air temperatures are simultaneously enhancing atmospheric demand for water vapor and precipitation[3]. This warming-induced intensification of the hydrological cycle is altering the availability of freshwater for growing crops at a time when population growth is demanding increases in food production and placing additional strain on water resources through industrial and household uses[4]. The United States Midwest, which supplies >30% of the world's corn (Zea mays) and soybean (Glycine max) grain[5], is especially vulnerable to changes in water availability as most of its agriculture is rainfed and strongly influenced by temperature[6,7]. With >1.5 °C of warming, global and regional climate models anticipate mixed effects on crop water availability, with projected stable or increasing precipitation, but also higher frequency and severity of droughts[8]. With more moderate warming, however, the impacts on Midwest hydrology are much less certain[9].

The amount of water used by crops is the sum of the daily evapotranspiration rate (ET; a total of soil evaporation and plant transpiration) from sowing until harvest. When a surface is well-supplied with water, ET is driven by the energy available to evaporate water. Hence, the potential evapotranspiration (PET) rate can be approximated via an energy balance, with the primary energy source being the net radiation reaching the surface and the sensible heat transferred by the wind from local or regional sources. Once crops reach full canopy cover (i.e., leaf area index >3) with sufficient soil water, the amount of water used by crops is equal to the PET[10–14]. The approach to calculate PET, first proposed by Penman[15] and later adapted by Monteith[16], is routinely used to estimate the atmospheric demand for water vapor and assess crop water needs[17]. PET is sensitive to the vapor pressure deficit (VPD) of the air[18,19], that is, the difference between saturation vapor pressure ($e_s$) and the actual vapor pressure ($e_a$). By definition, changes in VPD can only occur when $e_s$ and $e_a$ shift asymmetrically. Because $e_s$ increases exponentially with temperature (~7% per °C increase, according to the Clausius–Clapeyron relation), climate warming

in many regions could lead to raising VPD and PET, and thus increasing crop water needs[3,8].

Some studies have suggested that corn in the Midwest will require ~35% more water inputs to reach yield targets by 2050[20,21], arguing that sustaining yield gains will not be feasible without widespread expansion of irrigation infrastructure[22]. This claim is questionable, in part due to evidence showing that large differences in crop biomass and yield can be achieved with the same amount of ET, both within and across species with similar growing-season length[14,23]. But these studies also assume that climate warming in the Midwest is leading to increases in $e_s$ that are far outpacing increases in $e_a$. On the contrary, global daily minimum temperature ($T_n$), which in humid climates is a reasonable approximation of the dew point temperature ($T_{dew}$)[18], is rising 1.4 times faster than the daily maximum temperatures ($T_x$)[24]. Midwestern summer climate, specifically, has seen a long-term decline in $T_x$ and increasing precipitation has been observed despite moderate warming of daily average temperatures ($T_a$)[25–27]. However, it is unclear how these historical temperature trends have affected growing-season VPD and PET in the Midwest, and how future changes in atmospheric water demand under the persistence of these trends could impact crop yields.

Here, we show that potential crop water demand has remained unchanged despite the warming climate. This finding is based on an examination of climatological trends derived from near-surface meteorological observations at 1356 stations located throughout the Midwest, with records that extended as far back as 1894 (Fig. 1a). The null hypothesis of absence of monotonic trend was tested locally and regionally for aggregated summer climate (June, July, and August; JJA) variables as three time series: 1894–2019 (full record), 1960–2019 (60 years), and 1990–2019 (30 years). Examining the climate record as time series with differing lengths allowed us to check the robustness of the computed trends, as well as to better characterize recent climatic changes (see "Methods" for details about time-series selection). We then projected the temperature trends into 2050 to calculate the expected PET changes in the region under current climate trajectories and predicted the potential impacts on corn water stress and yields with ancillary use of the SALUS (systems approach to land-use sustainability) crop model[28,29].

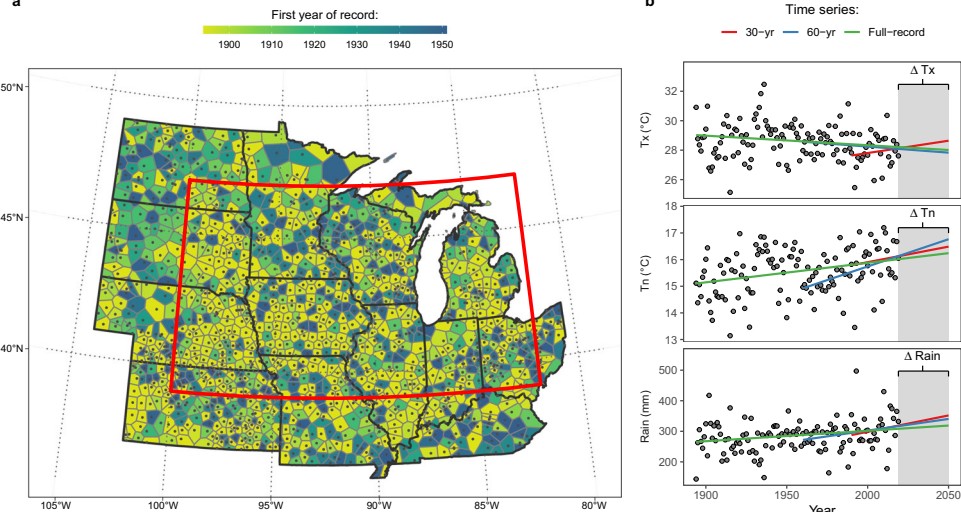

**Fig. 1 Weather data and analytical approach. a** Location of weather stations used. Daily observations for the summer growing season (JJA) were available from 1894 to 2019 for 485 stations, and 1951 to 2019 for all 1356 stations. Voronoi polygons indicate station spatial coverage. Red rectangle indicates the study area. **b** Example of the approach used to derive the historical summer climate trends and projections into 2050. Points indicate the average value across all weather stations. Lines indicate computed trends based on Sen's slopes for three time series.

## Results

**Regional and local summer climate trends**. Analysis of the temperature records in our dataset found no consistent overarching regional trend for $T_a$ ($P > 0.1$) for either of the three time series (Table 1), with most of the study area (>60%) exhibiting no systematic local trend in $T_a$. Those stations that did show a trend mainly cluster in eastern states (Indiana, Ohio, Illinois, Michigan, and Wisconsin; Fig. 2), resulting in warming at a rate 0.15–0.25 °C per decade. Meanwhile, mean summer $T_n$ did show a generalized warming trend, with 62–75% of the study area seeing significant increases in $T_n$ for both the full-record and 60-year trends, respectively (Table 1). Decadal increases for the full record were of much lower magnitude, indicating accelerated warming after 1960. These increases in $T_n$ agreed with similar regional increases in mean summer $T_{dew}$ in both the full-record and 60-year time series, at a rate that was 1.3 times the nominal increase in $T_a$, but about half the rate of increase in $T_n$ (Table 1).

On the other hand, an overall regional cooling trend in mean summer $T_x$ was detected at a rate of 0.062 °C decade$^{-1}$, but only for the full-record series ($P < 0.1$; Table 1). Although the 60-year cooling trend in $T_x$ was not generalizable to the whole region, it did exhibit a spatial gradient, with the majority of stations with a strong cooling trend clustering within the north–western quadrant of the study region (i.e., Iowa, Nebraska, South Dakota, and Minnesota; Fig. 2). This suggests that the decline in $T_x$ in the western plains is driven by more recent changes, whereas it may have softened in the eastern portion of the study region. It is worth noting that we did not find significant regional trends in any of the temperature variables for the 30-year time series, which is consistent with most stations (83–91%) not showing significant local trends. This points to the potential inadequacy of short-term climate records (<30 years) for detecting temperature trends.

Based on these historical temperature records, we estimated widespread increases in air-specific humidity (SH; unit mass of water vapor per unit mass of air) for the full-record and 60-year time series (Table 1 and Fig. 3). These increasing humidity trends matched the observed regional increases in summer rainfall of 7.7 mm per decade during the 60-year time series and with local trends up to 20 mm per decade in some areas. Again, the increases in SH and rainfall are about twice the long-term trend estimated for the full record (3.2 mm per decade), which is consistent with the acceleration of the warming trend in $T_n$ and $T_{dew}$ during the last 60 years.

Despite these generalized warming trends in $T_n$ and more localized increases in $T_a$, we were only able to detect significant regional trends in VPD in the full-record time series (Table 1 and Fig. 3). In fact, spatial patterns in VPD changes resemble those of $T_x$ (cf. Figs. 2 and 3), and the central and western portions have decreased in mean summertime VPD at a rate between 0.01 and 0.04 kPa per decade since 1960, which translates to decreases in cumulative JJA PET of up to 6 to 8 mm per decade (Fig. 3).

**Projected changes in evaporative demand by mid-century**. We used computed decadal trends in $T_x$, $T_n$, and $T_{dew}$ extrapolated to 2050 to estimate how growing-season evaporative demand could change, assuming that solar radiation ($R_s$) and wind speed ($U$) remain unchanged (see Methods). A comparison among the projected ΔPET for the three times series is shown in Fig. 4 (see also Supplementary Fig. 1 for ΔVPD comparisons). Based on these projections, we expect the region to experience little change in evaporative demand, with summer average daily VPD and PET increasing, respectively, by 1.4% and 0.6%, if the three time series are averaged. Consistent with the historical record, there is a

| Variable | Time series | Regional trend# | Local trend | | |
|---|---|---|---|---|---|
| | | | Increasing | Decreasing | No trend$ |
| | | (Δ decade$^{-1}$) | Share of area (%) | | |
| $T_a$ (°C) | 1894–2019 | 0.025 | 25 | 16 | 60 |
| | 1960–2019 | 0.072 | 33 | 6 | 61 |
| | 1990–2019 | 0.17 | 16 | 1 | 83 |
| $T_x$ (°C) | 1894–2019 | **−0.062**** | 8 | 52 | 39 |
| | 1960–2019 | −0.079 | 7 | 37 | 56 |
| | 1990–2019 | 0.16 | 13 | 3 | 84 |
| $T_n$ (°C) | 1894–2019 | **0.068**** | 62 | 6 | 31 |
| | 1960–2019 | **0.21***** | 75 | 1 | 24 |
| | 1990–2019 | 0.17 | 22 | 3 | 75 |
| $T_{dew}$ (°C) | 1894–2019 | **0.043***** | 75 | 5 | 21 |
| | 1960–2019 | **0.097***** | 74 | 1 | 25 |
| | 1990–2019 | 0.071 | 22 | 4 | 74 |
| Rain (mm) | 1894–2019 | **3.3**** | 41 | 0 | 59 |
| | 1960–2019 | **7.7*** | 25 | 1 | 75 |
| | 1990–2019 | 8.1 | 5 | 1 | 94 |
| SH (g/kg) | 1894–2019 | **0.032***** | 75 | 5 | 21 |
| | 1960–2019 | **0.07***** | 74 | 1 | 25 |
| | 1990–2019 | 0.053 | 22 | 4 | 74 |
| VPD (kPa) | 1894–2019 | **−0.0067*** | 9 | 49 | 42 |
| | 1960–2019 | −0.0073 | 8 | 34 | 58 |
| | 1990–2019 | 0.018 | 14 | 3 | 83 |
| PET (mm) | 1894–2019 | **−1.0*** | 9 | 45 | 45 |
| | 1960–2019 | −0.84 | 9 | 28 | 63 |
| | 1990–2019 | 3.2 | 14 | 2 | 84 |

**Table 1 Summary of regional and local trends for the three time series.**

Response variables: summer daily average ($T_a$), maximum ($T_x$), minimum ($T_n$), and dew point ($T_{dew}$). Temperatures, specific humidity (SH), rain, vapor pressure deficit (VPD), and potential evapotranspiration (PET).
#Regional trend significance codes in bold: *: $0.1 > P \geq 0.05$, **: $0.05 > P \geq 0.01$, ***: $0.01 > P \geq 0.001$; ****: $P < 0.001$.
$Nonsignificant local trend at the $\alpha = 0.1$ level.

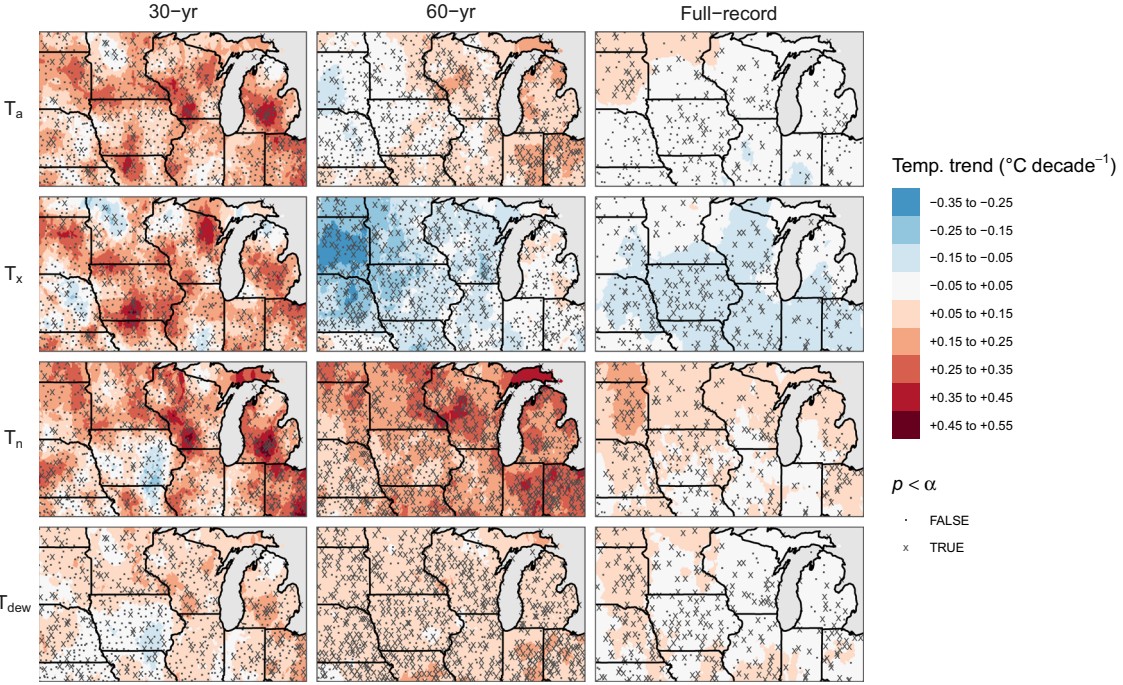

**Fig. 2 Spatial patterns of local trends in mean summer daily average ($T_a$), maximum ($T_x$), minimum ($T_n$), and dew point ($T_{dew}$) temperatures for the three time series.** Stations marked with a cross ($x$) indicate significant local trends at the $\alpha = 0.1$ level.

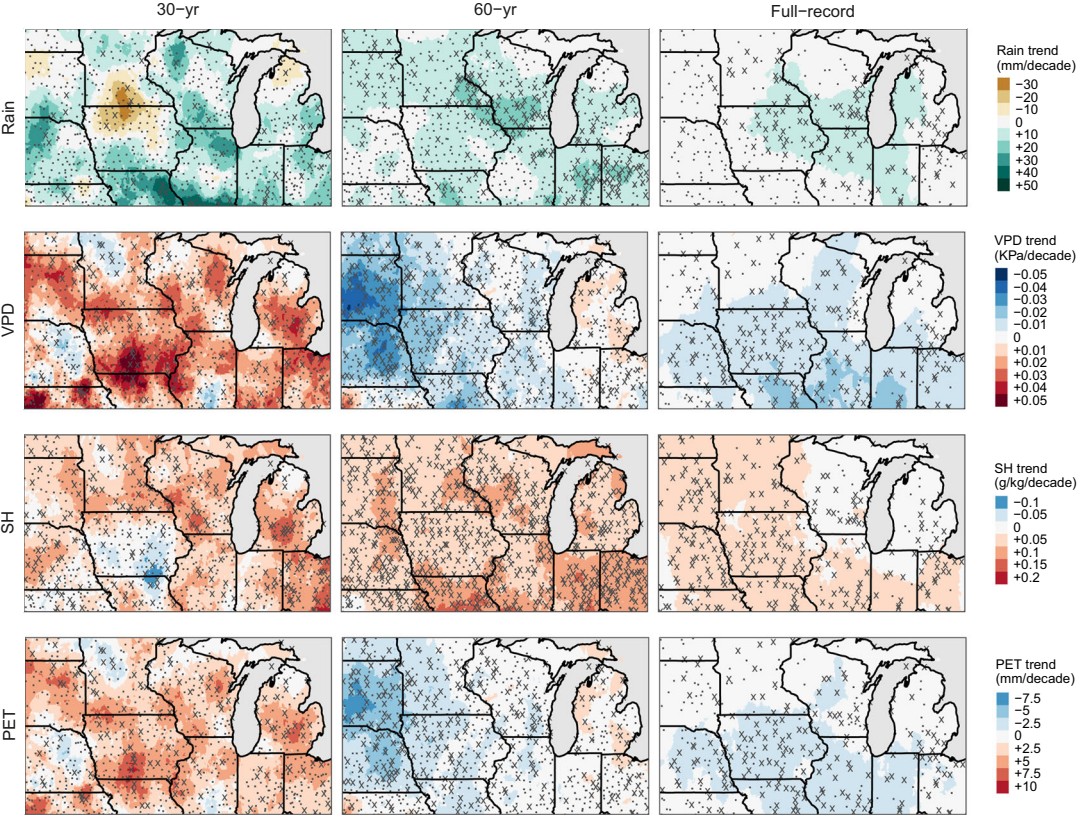

**Fig. 3 Spatial patterns of local trends in mean summer (JJA) rain, specific humidity (SH), vapor pressure deficit (VPD), and total summer potential evapotranspiration (PET) for the three time series.** Stations marked with a cross ($x$) indicate significant local trends at the $\alpha = 0.1$ level.

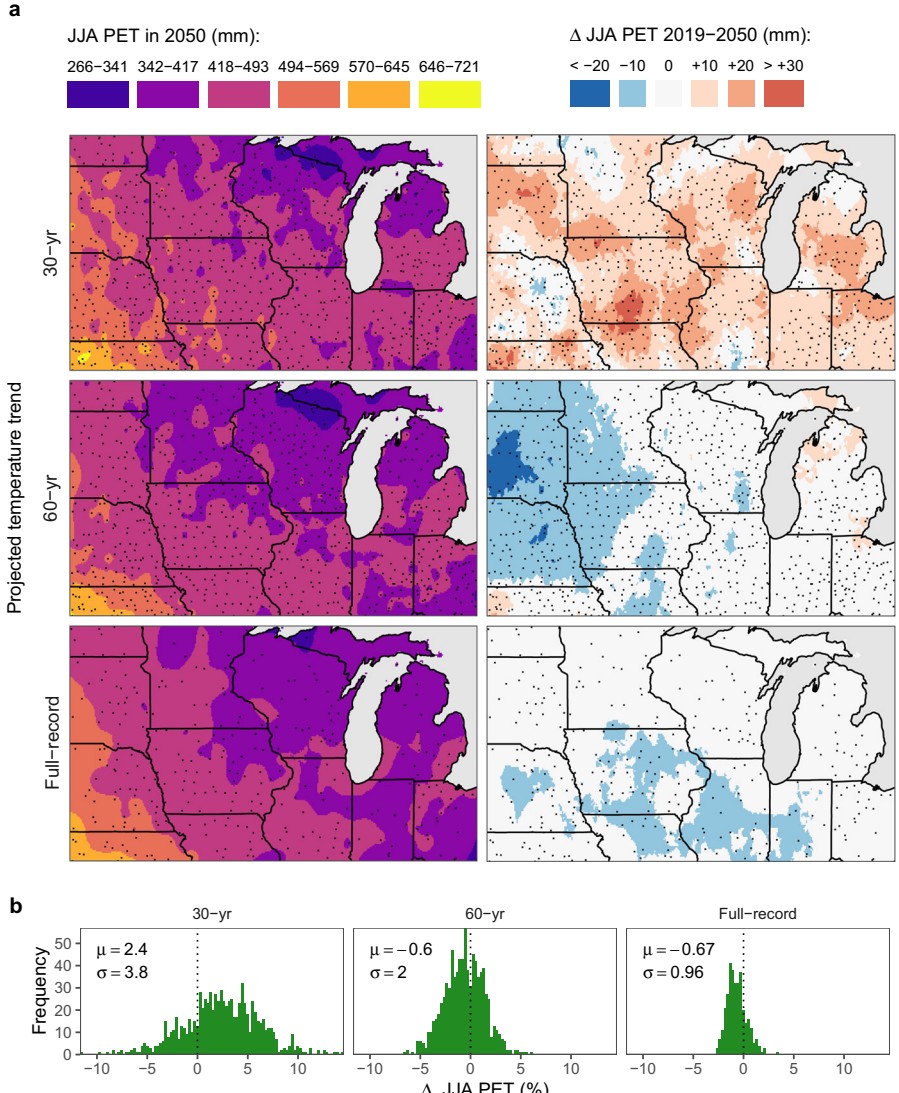

**Fig. 4 Potential evapotranspiration (PET) based on temperature trends for 30-year, 60-year, and full-record times series projected into 2050.**
**a** Spatial patterns of projected PET and their absolute change compared to current climate. Points indicate locations of the weather stations. **b** Histograms depict the distribution of the projected PET (as relative percent change) across all weather stations.

disagreement between the direction of the ΔPET change, with both the 60-year and full-record trend projections indicating a slight decrease (2.7–2.9 mm or 0.6–0.7% on average) and the 30-year trend projection indicating increasing PET (10.6 mm or 2.5% on average; Fig. 4b). The same pattern is shown for ΔVPD, though the relative changes are more substantial (Supplementary Fig. 1).

It is worth noting, however, that the 30-year time series projections do have a greater degree of uncertainty, as reflected in the widespread lack of significance in local trends and no overarching regional trends for any of the variables examined (Table 1). In fact, approximating the uncertainty in our VPD projections given the standard errors of the estimated temperature trends shows that the propagated uncertainty for the 30-year time-series projection (11–32%) is about 2.5-fold that of its 60-year counterpart (5.9–13%), and about fivefold that of the full-record projection (3.2–6.3%; see Supplementary Fig. 2).

**Impacts of projected climate on corn water stress and yield**. We next explored the potential impact of projected PET and rainfall changes on crop growth by simulating corn growth under

historical and 2050 climate scenarios for the three time-series trends. For this, we used SALUS, a process-based crop model which has been shown to be capable of capturing soil hydrology and evapotranspiration dynamics across many crops and soils[10,14,29]. SALUS estimates daily PET via an energy balance, derived from Penman's equations approach[23]. We assessed the adequacy of the model for predicting the yield responses to climatic changes, first by checking the plausibility of simulated yields under the historical weather. These showed good agreement with the state-aggregated yields reported by the National Agricultural Statistics Service (NASS)[30] for the 2010–2019 period (Supplementary Fig. 3). Second, we examined the simulated ΔPET, which matched closely to that estimated by Penman–Monteith ($r^2 = 0.94$; Supplementary Fig. 4). Likewise, the simulated ΔPET map closely resembles projections reached by Penman–Monteith (c.f. Fig. 4 and Supplementary Fig. 5), with 30-year trend simulations showing a positive change (8.0 mm) and the 60-year and a full record showing a negative change (9.7 and 6.1 mm, respectively).

The spatial patterns of simulated changes in corn yield and water-stress days for stations located in corn-producing counties

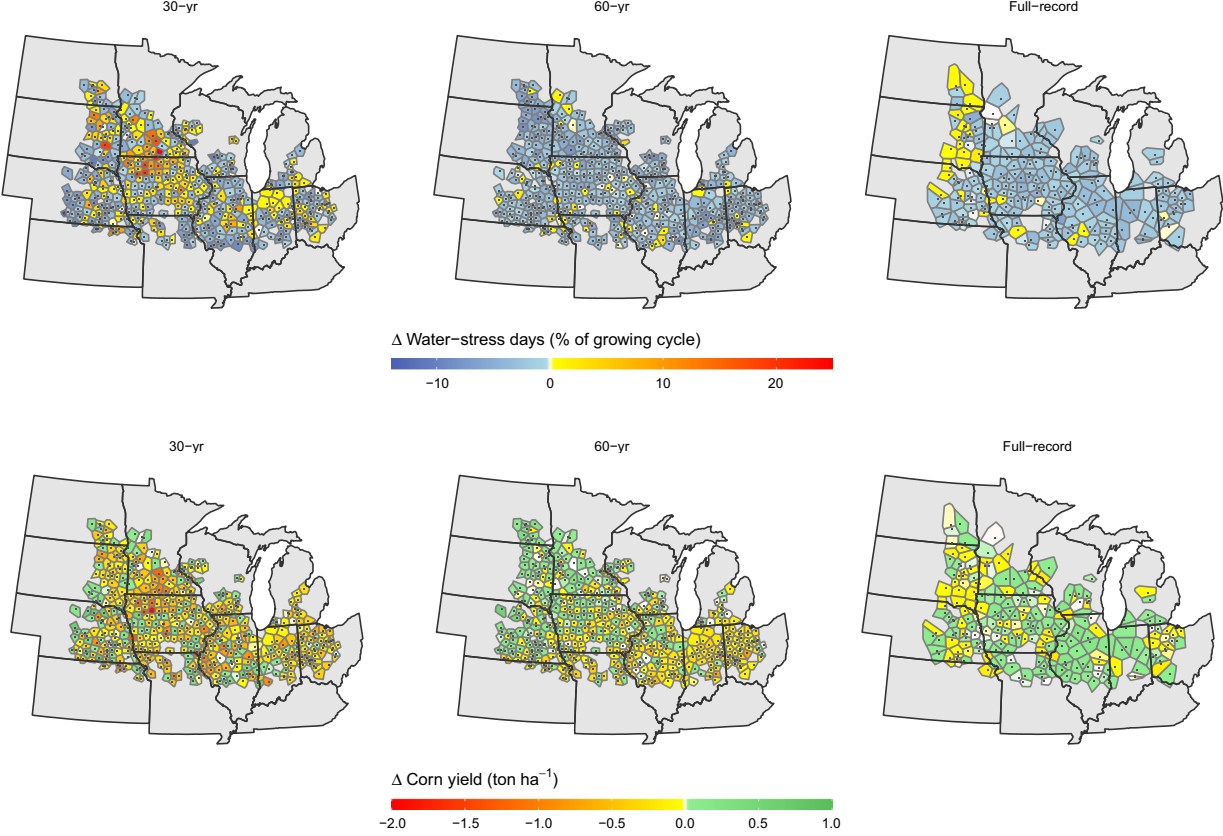

**Fig. 5 Impacts of projected summer temperature and rainfall trends on corn water-stress days and yield as simulated by the SALUS model.** Voronoi polygons indicate coverage of stations within major corn-producing Midwestern counties.

are shown in Fig. 5. Simulation under all three projections scenarios predicts slight decreases in the proportion of days in which corn experiences water stress, from an average of 21.9% of the growing season under current weather to 20.6% under the weather projections. Corn yield is predicted to decrease on average by 1.7% (0.19 ton ha$^{-1}$) for the 30-year trend projection, whereas a change of less than half a percent is predicted for the 60-year and full-record projection scenarios (Fig. 5).

As shown in Fig. 6, the simulated responses in corn water stress and yields reflect changes in both JJA PET and rainfall. Yet, simulated changes in water-stress days (expressed as the percent of growing cycle length) are more strongly correlated with changes in total rainfall than corn yield; a 100 mm increase in rainfall is associated with an average decrease of 6.0 water-stress days ($r^2 = 0.50$). On the other hand, yield shows a stronger correlation to changes in PET ($r^2 = 0.37$); a 10 mm increase in PET is associated with an average decrease of 0.1 tons ha$^{-1}$ in corn yield. Only ~10% of simulations were predicted to experience both increases in PET and decreases in rainfall (i.e., aridification).

## Discussion

Climate change is disrupting crop water supply and demand in many of the world's agricultural regions, but recent climatic trends in the Midwest have likely been beneficial for crop water availability[7,31]. Our analysis demonstrates that the contrasting trends in $T_{dew}$ and $T_x$ have had important implications for atmospheric water demand and crop water use. If both $T_{dew}$ and $T_x$ were increasing at roughly the same rate, we would expect increases in VPD and therefore increased PET. In actuality, the extent to which growing-season $T_a$ warming is occurring in the

Midwest seems largely driven by increases in $T_n$ that are in turn matched by similar increases in $T_{dew}$, albeit of smaller magnitude (Table 1).

The concomitant increases in $T_{dew}$ and decreases in $T_x$ have resulted in either stable or declining PET throughout much of the Midwest (Fig. 3). Rising $T_{dew}$ reflects increases in air-specific humidity, which are generally linked to global increases in oceanic evaporation and other global hydrologic cycling patterns[32,33]. Meanwhile, the long-term cooling of summertime $T_x$ is consistent with the so-called "warming hole" phenomenon in the Midwest[25,34], which has persisted over the eastern United States since the late 1950s[26]. Though little consensus exists on its definite causes, likely contributing factors include oceanic oscillations[26,35,36], increased cloud cover[37], anthropogenic aerosols[38], and regional agricultural intensification[39,40].

A body of evidence has emerged in support of the latter explanation[39–42], with the hypothesized link being an evaporative cooling effect (i.e., latent heat) as a consequence of elevated rates of ET due to denser canopies[11,40] and increased irrigation[43–46]. Agricultural management changes throughout the second half of the 20th century, such as increased plant populations, earlier planting, and fertilizer additions, have indeed resulted in prolonged periods with full canopy closure (i.e., leaf area index >3) at which ET would be expected to be near to PET if sufficient soil moisture is available[11]. Therefore, the period of peak ET has likely widened, providing the basis for the increased evaporative cooling effect. On the other hand, because irrigation in the Midwest is mainly confined to the cropland west of the 96th meridian (i.e., Nebraska and South Dakota[47]), we would expect irrigation to have a limited impact. Yet, it is clear that these irrigated areas in fact exhibited the

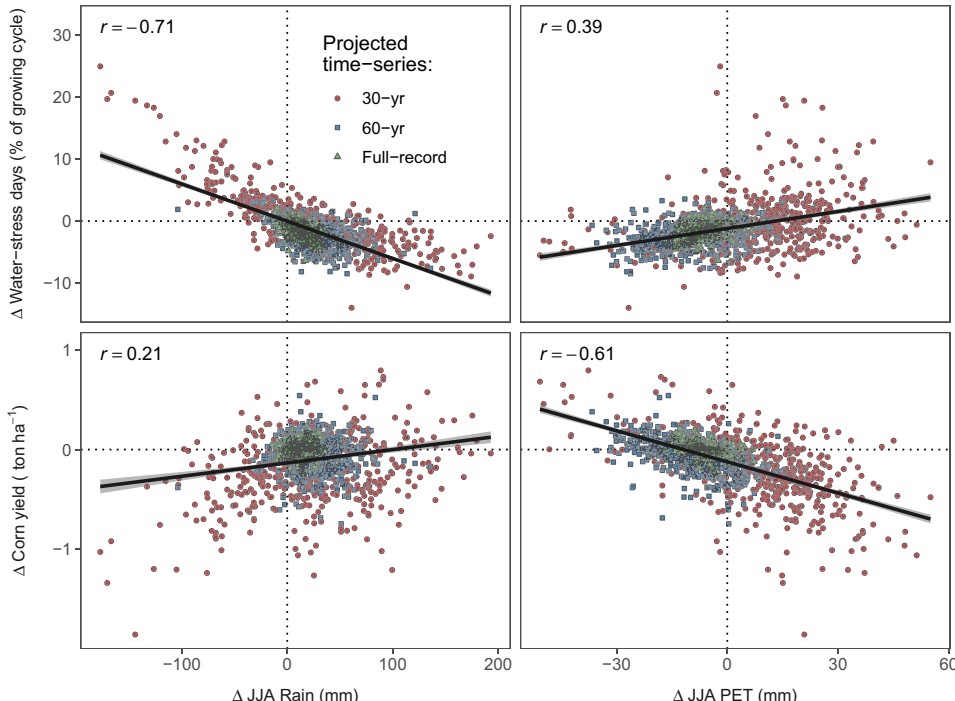

**Fig. 6 Correlation of changes in summer rainfall and potential evapotranspiration (PET) to changes in corn water-stress days and yields under projected climatic changes.** Each symbol indicates the average value of the simulation at a weather station with a given time series. $r$ = coefficient of correlation.

strongest negative VPD and PET trends, particularly in the 60- and 30-year time series (Fig. 3), which is also consistent with such ET-cooling mechanism.

This decrease in evaporative demand has taken place simultaneously with the lengthening of the growing seasons[7] and increasing precipitation (Fig. 3)[27], favoring water surpluses during the period of active crop growth. Barring changes in the intensity and timing of rainfall[27,48], this suggests that yield loss due to water deficits has, on average, become less frequent in the region. Indeed, a previous study estimated that 28% of the corn yield gain in the Midwest since 1981 is attributable to better weather, of which summer $T_x$ cooling accounted for almost half of this beneficial trend[7]. Another study[37] that examined the counterfactual, estimated that warming-hole climate boosted corn yields by ~5–10%, mainly due to prolonged maturity time and reduction of water stress[31].

Based on our temperature projections of the full-record and 60-year time series, we estimate summertime PET in the region could be expected to decrease by 0.6–0.7% on average by 2050 (Fig. 4). According to our simulations with the SALUS crop model, which were able to replicate satisfactorily our projected ΔPET estimated offline by the Penman–Monteith method (Supplementary Fig. 4), these projected changes in PET and precipitation could largely reduce corn water deficits and have a neutral to a slightly beneficial effect on corn yields (Fig. 5). Though we must point out that our delta approach, which focuses on the trajectory of the mean, does not account for changes in variability. Thus, yield impacts due to changes in the timing of precipitation or increased frequency of drought that could affect sensitive periods such as pollination or kernel set, which can occur despite decreases in mean PET[49,50], are not considered.

Our projections of decreasing PET contrast with the projected increases in VPD for the Midwest based on simulations from general circulation models[21,51]. Though it should be noted that projections from CMIP5 and earlier models have been shown to have limited skill at recreating the warming-hole trends observed in the Midwest[34,35,40]. Also, the degree of change in VPD projections largely depends on future emission scenarios. For instance, one study[35] projected the likely disappearance of the warming-hole trends during the twenty-first century under a high-emissions scenario (RCP 8.5), whereas showing some chance of persistence under a climate stabilization scenario (RCP 4.5). Indeed, there is some evidence suggesting that the $T_x$ cooling trend has weakened since the 1990s[36], which is in agreement with the nominal 30-year trends found in our dataset (Table 1 and Fig. 3). It is also possible that the reversal of the $T_x$ warming trend in the 30-year time series found here could be an artifact, possibly because the 1990s was the decade with the lowest $T_x$ on record (Supplementary Fig. 6), and compounded by the limited number of years available to compute reliable trends (see lack of significance of 30-year trends on Table 1). It is worth noting that although $T_x$ did increase in subsequent decades, $T_x$ was still cooler than the full-record average during 1990–2019 (Supplementary Fig. 6).

The discrepancy among time-series trends exposes the inherent limitations of ΔPET projections made from historical records. Hence, these should be interpreted with caution given the incomplete understanding of the mechanisms driving the $T_x$ cooling trend, and the possibility that the trajectory of decreasing evaporative water demand could subside in the future[36], particularly under high emission scenarios[35]. Other uncertainties stem from possible decreases in $R_s$ resulting from increasing cloud cover and aerosols[37,40], which were not considered in our analysis but would likely result in lower $R_s$ and thus decrease PET even further. Nevertheless, it is noteworthy that even in the cases when temperature trends do lead to increases in PET in our simulations, impacts on corn yields should remain small. For example, even the upper boundary of the projected ΔPET (55 mm; Fig. 6) is predicted to result in an average yield loss of 0.7 Mg ha$^{-1}$ (or 6.1%).

These findings, along with our previous work demonstrating that achieving yields three times greater than current averages are possible with roughly the same amount of ET[11,23], calls into question claims about the need to aggressively expand irrigation in the rainfed Midwest as a response to climate change. A recent study[22], for example, estimated based on CMIP5 model projections that up to 260 mm of additional water would be needed by mid-century to support current yield gain trends in the Midwest, which would require a three- to sixfold expansion in the irrigated area. Yet these water input levels are 6–15 times greater than the average ΔPET projected here based on the 30-year trend (Fig. 6). Irrigation is indeed a valuable tool for managing risks and boosting profits and will continue to play an important role in adapting production systems to increased weather variability under climate change. Still, unless there is an acceleration of warming that is orders of magnitude greater than the historical trends, we expect that, on average, increases in summer rainfall will be sufficient to sustain yields despite the persistence of growing-season $T_a$ warming trends. We contend that this fact has largely been overlooked when discussing ongoing and future impacts of climate change on Midwestern agricultural production. Further research should focus on elucidating how long-term changes in PET and precipitation in the region have influenced management and genetics and the inherent vulnerabilities that they may carry into future climate scenarios.

## Methods

**Weather station records.** We acquired daily $T_x$, $T_n$, and precipitation observations for the summertime months of JJA for 1356 stations across 13 Midwestern states (Fig. 1a). Stations are part of the National Weather Service Cooperative Observer Program (NWS-COOP). We retrieved the entirety of the weather record (up to 2019), with observations as early as 1894 in 485 stations. All stations included observations from at least 1951.

Because the $T_{dew}$ for the NWS-COOP stations were not reported, we estimated $T_{dew}$ using a Generalized Additive Model (GAM) approach. The GAM was trained with data from the Automated Surface Observing System (ASOS) network stations within the study region ($n = 462$; Supplementary Fig. 7). Average growing season $T_{dew}$ was predicted as a function of smoothing effects of $T_n$, $T_x$, and the 2D smoothing effect of longitude and latitude (Supplementary Fig. 8). The developed GAM was able to satisfactorily reproduce $T_{dew}$ ($r^2 = 0.79$; Supplementary Fig. 9) in independent ASOS stations ($n = 198$), which was better than using $T_n$ as a surrogate for $T_{dew}$ ($r^2 = 0.67$; Supplementary Fig. 9).

Weather data for both NWS-COOP and ASOS stations were accessed through the Iowa Environmental Mesonet (IEM)[52], which provides high-quality daily weather summaries of temperature and precipitation records with minimal or no missing data. Details about quality control procedures can be seen in the IEM website (https://mesonet.agron.iastate.edu/). In addition to the protocols performed by IEM, we conducted our own manual quality control which included: (i) removing unreasonable values that would indicate sensor malfunction (e.g., temperatures < −45 or >50, precipitation events >500 mm); and (ii) replacing missing values (<0.1% of the record) with the previous day value, unless there were more than 4 consecutive days of missing data, in which case we excluded that year from the analysis.

**Calculation of VPD and PET.** The VPD was calculated as the difference between the average of the saturated vapor pressure ($e_s$) at $T_x$ and $T_n$ (i.e., daily average $e_s$), and the actual vapor pressure of the air, which was approximated as the $es$ at the $T_{dew}$

$$\text{VPD} = \frac{es(T_x) + es(T_n)}{2} - es(T_{dew}) \qquad (1)$$

Saturated vapor pressure (kPa) at given temperature $T$ was derived using the following equation:

$$es(T) = 0.6108 * \exp\left(\frac{17.27 * T}{T + 237.3}\right) \qquad (2)$$

We calculated PET (mm day$^{-1}$) using the FAO Penman–Monteith equation for reference evapotranspiration for a well-watered grass of 0.12 m height, with fixed surface resistance of 70 s m$^{-1}$ and albedo of 0.23 for green crop surfaces[17]:

$$\text{PET} = \frac{0.408 * m * R_n + g * \left(\frac{900}{T_a + 273.3}\right) * U * \text{VPD}}{m + g * (1 + 0.34 * U)} \qquad (3)$$

where $m$ is the slope of the saturation vapor pressure curve at the mean daily temperature (kPa K$^{-1}$) calculated as $m = 4098 * es(T_a) * (T_a + 237.3)^{-2}$; $g$ is the psychrometric constant (kPa K$^{-1}$); U is the wind speed (m s$^{-1}$); and $R_n$ is the net irradiance (MJ m$^{-2}$ day$^{-1}$).

$$R_n = R_s * (1 - 0.23) - R_{nl} \qquad (4)$$

$R_s$ is the incoming solar shortwave radiation, and $R_n$ is the net shortwave radiation

$$R_n = \sigma * \left(\frac{(T_x + 273.3)^4 + (T_n + 273.3)^4}{2}\right) * \left(0.34 - 0.14\sqrt{es(T_{dew})}\right)$$
$$* \left(1.35 \frac{R_s}{R_{so}} - 0.35\right) \qquad (5)$$

where $R_{so}$ is the clear day $R_s$, which was approximated as:

$$R_{so} = (31.59 - 0.265 * \text{Lat}) + (0.8315 + 0.2417 * \text{Lat}) * \cos(0.0172 * (\text{DOY} - 170)) \qquad (6)$$

In this expression, Lat is the latitude (in decimal degrees) and DOY is the day of the year. Because daily $R_s$ and $U$ measurements were not available for the NWS-COOP stations, we used climatological averages derived from the gridded Prediction of Worldwide Energy Resources (POWER) dataset[53] (0.5° resolution), extracted from grid points at each station location.

For the psychrometric constant ($g$), we used the following formula:

$$g = 0.000665 * P \qquad (7)$$

where $P$ is the average atmospheric pressure (kPa) calculated as:

$$P = Pb * \left[\frac{Tb + Lb * h}{Tb}\right]^{\frac{-g_0 * M}{R * Lb}} \qquad (8)$$

where $Pb$ is the static pressure (101.3 kPa); $Tb$ is the standard temperature (293 K), $Lb$ is the standard temperature lapse rate (−0.0065 K m$^{-1}$); $h$ is the elevation above sea level (m); $g_0$ is the mean gravitational acceleration (9.80665 m s$^{-2}$); $M$ is the molar mass of Earth's air (0.0289644 kg mol$^{-1}$); and $R$ is the universal gas constant (8.31446 J mol$^{-1}$ K$^{-1}$). Elevation above sea level was extracted from data collected by the Shuttle Radar Topography Mission[54].

In addition to PET and VPD, we calculated the specific humidity of the air (SH; g kg$^{-1}$) using the following equation:

$$\text{SH} = \frac{1000 * 0.622 * es(T_{dew})}{P - (1 - 0.622) * es(T_{dew})} \qquad (9)$$

All variables were aggregated by year, so they represented the average for the JJA period at each station, given that preliminary analysis using daily values gave similar results with a small standard deviation.

**Time-series selection.** Selection of appropriate period for trend analysis is crucial given the long-term nature of the weather record available. We followed convention and set 30 years (1990–2019) as the minimum data required to derive meaningful climate trends, and then explored how trends changed by including additional data. To do this we computed Sen's slopes (see details below) for the observed weather station variables ($T_n$, $T_x$, and rainfall), and then worked backward in time until encompassing the whole record (1894–2019). This preliminary step revealed that slopes of both $T_n$ and rainfall were directionally consistent across the whole record, although their magnitude generally increased under more recent climate (Supplementary Fig. 10). On the other hand, $T_x$ maintained negative slopes of consistent magnitude until the late 1980's when the direction of the trend abruptly reversed. Accordingly, we used the 60 years as an intermediate length, as well as the full record (126 years). We hoped that the 60-year trend would capture the warming-hole period as shown in previous studies[25–27].

**Analysis of regional and local trends.** The dataset was inspected for missing data, and if a station did not contain all the years in the specified period, then it was discarded from the analysis.

The time series were subjected to a Mann–Kendall trend analysis trend test[55] to examine whether each of the studied variables aggregated for the summer period changed monotonically with a year as the predictor variable. We first tested a regional trend following the approach outlined by Helsel and Frans[56] with adjustment to account for correlation among sites. Then, we followed with the Mann–Kendall test at each location. The magnitude of detected trends was calculated following Sen's slope procedure[57], where the rate of change is the median of all linear slopes among observations within a dataset. The calculated annual Sen's slopes were multiplied by a factor of 10 to report them as decadal changes (Δ°C decade$^{-1}$). All procedures were implemented independently for each climate variable at each station, with a significance level of $\alpha = 0.1$.

The above methods are nonparametric and widely used to examine trends in temperature and rainfall due to their robustness against outliers in time-series data. They do, however, assume no autocorrelation in the time series. Because observations are far enough apart in time (1 year), autocorrelation was unlikely to be important in our dataset. This was confirmed by the general lack of lag-1 autocorrelation for each time series (Supplementary Fig. 11).

**Trend projections**. Spatial coverage of significant trends was calculated using Voronoi polygons (e.g., Fig. 1a). The derived Sen's slopes from the trend analysis at each weather station were interpolated using ordinary kriging to examine spatial patterns in temperature trends for each time series. The interpolated surface was subset to a boundary within latitudes 39–47°N and 82 and 100°W (Fig. 1a), which encompasses the majority of cropland in the US Midwest. The historical temperature trends were extrapolated observations from 2019 to 2050 and used to compute future changes in VPD and PET. $R_s$ and $U$ were assumed to remain unchanged.

**Crop growth simulations**. We used the SALUS model[28,29] to quantify the mechanistic effect of the projected JJA temperature and precipitation trends into 2050. This model calculates the soil water balance on a daily time-step based on equations from the earlier CERES[58] models, with some differences in the calculation of infiltration, drainage, evaporation, and runoff[28]. For ET, SALUS uses the procedures outlined by Ritchie[13] with some modifications[59,60] to calculate the soil evaporation and plant transpiration terms of ET independently, both as a function of leaf area index (LAI) and the radiation and aerodynamic components of the energy balance. Details about the calculation of PET in SALUS are provided by Basso and Ritchie[23].

Simulations were conducted at every station, excluding those within counties without significant corn production (i.e., <5000 ha of corn harvested on average during the last 10 years). For the baseline, we use the historical record for 1984–2019 using daily $T_x$, $T_n$, and precipitation recorded at the weather stations, and solar radiation (MJ m$^{-2}$) from the corresponding grid cells within the NASA Power dataset[53]. Future weather data were generated using the delta method[31] based on the temperatures and precipitation trends for each time series, projected into 2050. The projected climate variables were then used to derive an additive factor for $T_x$ and $T_n$, and a multiplicative factor for precipitation, which was applied only during the JJA period. Radiation records were not changed.

We used data from the soil survey geographical (SSURGO) database[61] to select the predominant soils within a 40-km search radius from each station. Field management followed typical practices for the region, with planting and harvest dates based on survey reports by NASS[30]. We also assumed moderate tillage regimes and sufficient nutrient levels. Because our objective was to quantify the effect of temperature trends on evaporative demand and crop water stress, all simulations assume rainfed conditions.

**Software**. Data processing and analysis were conducted in R (version 3.5.2) extended with the following packages: *trend* and *rkt* for local and regional trend analyses; *mgcv* for GAM fitting; *raster* and *SDraw* for spatial analysis and data processing; *tidyverse* for all other data wrangling; *rnaturalearthdata* for map data, and *ggplot2* for data visualization. Ordinary kriging interpolation was performed using the ArcGIS software (version 10.6, ESRI).

## Data availability
The authors declare that the main data supporting the findings of this study are available within the article and its Supplementary Information files. Extra data are available from the corresponding author upon request.

## Code availability
Code developed for this analysis is available from the corresponding author upon request.

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

## Acknowledgements

This research was supported in part by the Agriculture and Food Research Initiative—Education and Workforce Development program from the US Department of Agriculture National Institute of Food and Agriculture award numbers: 2020-67021-32799, 2019-67012-29595, 2018-67003-27406, 2017-67013-26257, and by the Great Lakes Bioenergy Research Center, US Department of Energy, Office of Science, Office of Biological and Environmental Research under award numbers DE-SC0018409 and DE-FC02-07ER64494. We acknowledge the Iowa Environmental Mesonet for archiving, maintaining, and providing access to high-quality meteorological data, and the funding agencies and volunteers that support this effort.

## Author contributions

B.B. and J.T.R. conceived the research question, designed the study, and supervised the research; J.T.R. devised the methodology for calculating VPD and PET; L.R. gathered, organized, and curated the meteorological data. R.A.M.-F. and L.R. performed trend analyses, climate projections, and crop simulations; R.A.M.-F., L.R., and B.B. conceptualized and created the figures; B.B., R.A.M.-F., and J.T.R. interpreted the results. R.A.M.-F. and B.B. wrote the paper.

## Competing interests

The authors declare no competing interests.
