## [Peer Review File · Nature Communications]

REVIEWER COMMENTS

Reviewer #1 (Remarks to the Author):

This paper examines trends in the temperature characteristics (t_{max} , t_{min} , dew point, etc.), VPD, specific humidity, and their effects on PET in the midwestern United States. They find, based on observations, that average air temperature has warmed largely from increases in t_{min} and t_{dew} , but stable/decreasing t_{max} . This translates to decreases in VPD and thus PET (or reference ET, which the authors use). Overall, the analysis seems sounds, but I do have issues, some of which are major.

1. My biggest issue is that throughout the paper the trends in the variables are largely based on the 1960-2019 trends. The 1990-2019 trends, however, really tell a different story. They suggest widespread increases in average temperature, maximum temperature, minimum temperature, and slight increases in dew point temp. Further, this time period shows large increases in VPD and widespread increases in PET, in contrast to the 1960-2019 time period. It seems, then, that the conclusions from this paper are largely due to the time period selection, which is discussed only briefly in the discussion section. The authors suggest that they didn't use the 1990-2019 time period because of autocorrelation issues, but Figure S4 doesn't suggest that to be a significant issue. Further, why did the authors choose 1990 as a starting point? The authors should think about going back to 1980, which most studies show is when the climate change signal really starts to leave natural variability. Overall, I would like for more discussion to be added on the time period selection and if their results are an artifact of the time period selected.
2. For the 2019-2050 trends, the authors do not state which time series was used for extrapolation in the text. I assume it was the 1960-2019 time period. If the authors use the 1990-2019 extrapolation then the projection maps will look quite different. The extrapolated maps from the different time could/should go into the supplemental.
3. Why didn't the authors use relative humidity observations to estimate e_a ? Where there not enough observations?
4. There have been many studies that have examined the effects of irrigation on near-surface climate, where irrigation has shown to cool surface temperatures and decrease VPD (because of increases to humidity). How does irrigation in this region affect the results? See citations below. I would like to see a little discussion on this added to the paper.
5. The supplemental information document needs more detailed information. The figure captions do not give enough information to understand what the figure contains.

Sacks, W. J., Cook, B. I., Buening, N., Levis, S., & Helkowski, J. H. (2009). Effects of global irrigation on the near-surface climate. *Climate Dynamics*, 33(2-3), 159-175.

Cook, B. I., Shukla, S. P., Puma, M. J., & Nazarenko, L. S. (2015). Irrigation as an historical climate forcing. *Climate Dynamics*, 44(5-6), 1715-1730.

Cook, B. I., McDermid, S. S., Puma, M. J., Williams, A. P., Seager, R., Kelley, M., ... & Aleinov, I. (2020). Divergent regional climate consequences of maintaining current irrigation rates in the 21st century. *Journal of Geophysical Research: Atmospheres*, 125(14), e2019JD031814.

Qian, Y., Huang, M., Yang, B., & Berg, L. K. (2013). A modeling study of irrigation effects on surface fluxes and land-air-cloud interactions in the Southern Great Plains. *Journal of Hydrometeorology*, 14(3), 700-721.

Reviewer #2 (Remarks to the Author):

Thank you for the opportunity to review this paper. It is a well written and well researched article that is suitable for publication in Nature Communications. The topic is of high importance as it discusses an implication of climate change as it relates to the highly productive agricultural region of the central U.S.

I recommend the article be published with consideration of the following revisions.

I feel there should be a short discussion on crop coefficients, which could be included in lines 40 – 50 of the manuscript. ET rates vary by crop type and vary significantly over the course of the growing season. Would there be any impact on the outcome of the research if these are factored?

Were there any quality control measures performed on the COOP and ASOS station data? The authors state the source of the data is the Iowa Environmental Mesonet program. Does this network offer quality control procedures? ASOS data in particular can be cumbersome to analyze as it includes sub-hourly data taken at what can be irregular timeframes. A brief discussion on quality of the dataset is warranted, in my opinion.

There are several assumptions made in the calculations. What sources of errors might there be in the assumptions used? Perhaps error bars on the PET estimations might be warranted. Rs and U were fixed and do not change in the future – is this a safe assumption to make?

Finally, it might be good to add a few more items to the discussion related to climate change implications for agriculture – namely potential of erosion from heavy rain events, timing of heat events in relation to corn pollination, lengthened growing season, etc.

Good luck with the publication process,
Martha Shulski

Reviewer #3 (Remarks to the Author):

In this MS, the authors investigate past long-term trends in temperature and future changes in potential evapotranspiration in the agriculture region of Midwest US. The paper presents a topic which is of great interest within the agricultural communities given the potentially significant changes in crop growth that could occur as a result of ongoing climate warming. The paper is very well written. However, this analysis should be extended to demonstrate the added value of the trend analysis beyond just using the methods for projecting changes in summer PET. I didn't find the results too intriguing since they essentially showed the impacts of contrasting trends in Tx and Tdew and how these trends would affect the growing season PET. I would have expected the authors to have carried out additional analysis of changes in the water deficit combined with trends in crop yield and come up with more scientific statements about how the crop growth could be affected. As currently presented, this work provides no useful information that can benefit framers in their attempt to cope with the projected changes due to climate change.

I would like to see at least some additional analysis of crop yield trends to also show how the changes in temperature and PET would influence the crop growth for understanding of past interactions between yield and climate trends but also the future changes. If the authors have more results in this vein, they should upgrade their study accordingly.

REVIEWERS' COMMENTS

Reviewer #1 (Remarks to the Author):

The authors did a considerable amount of work and I am satisfied with their responses.

Reviewer #2 (Remarks to the Author):

Thank you for addressing the comments in this revision. I support publication of the manuscript.

Reviewer #3 (Remarks to the Author):

The Authors have made significant modifications to the manuscript in order to address the concerns raised in the first round of review especially in regard to the impact of potential climate change on crop yield. I am satisfied by the updated version. I consider this article suitable for publication.

REVIEWER COMMENTS

Reviewer #1 (Remarks to the Author):

This paper examines trends in the temperature characteristics (t_{max} , t_{min} , dew point, etc.), VPD, specific humidity, and their effects on PET in the midwestern United States. They find, based on observations, that average air temperature has warmed largely from increases in t_{min} and t_{dew} , but stable/decreasing t_{max} . This translates to decreases in VPD and thus PET (or reference ET, which the authors use). Overall, the analysis seems sounds, but I do have issues, some of which are major.

1. My biggest issue is that throughout the paper the trends in the variables are largely based on the 1960-2019 trends. The 1990-2019 trends, however, really tell a different story. They suggest widespread increases in average temperature, maximum temperature, minimum temperature, and slight increases in dew point temp. Further, this time period shows large increases in VPD and widespread increases in PET, in contrast to the 1960-2019 time period. It seems, then, that the conclusions from this paper are largely due to the time period selection, which is discussed only briefly in the discussion section. The authors suggest that they didn't use the 1990-2019 time period because of autocorrelation issues, but Figure S4 doesn't suggest that to be a significant issue. Further, why did the authors choose 1990 as a starting point? The authors should think about going back to 1980, which most studies show is when the climate change signal really starts to leave natural variability.

Overall, I would like for more discussion to be added on the time period selection and if their results are an artifact of the time period selected.

Thank you for this comment.

Selecting a period for trend analysis is always tricky given that it is fundamentally an arbitrary decision. In this study we examined climate record as time-series with differing lengths to allow us to check the robustness of the computed trends, as well as to better characterize recent climatic changes. To select the three time-series, we started with the convention and set 30 years (1990-2019) as the minimum data required to derive meaningful climate trends and then explored how trends changed by including additional data, as shown in the figure below:

This preliminary step revealed that slopes of both Tn and rainfall were directionally consistent across the whole record, although their magnitude generally increased under more recent climate. On the other hand, Tx maintained negative slopes of consistent magnitude until the late 1980's when the direction of the trend abruptly reversed. Accordingly, we decided to use 60-years as an intermediate length, as well as the full record (126-yrs). We hoped that the 60-yr trend would capture the “warming-hole” period as shown in previous studies (e.g. Pan et al, 2004; Partridge et al, 2018; Dai et al, 2016).

That being said, we agree that the reversal of the Tmax trend the 30-yr trend could indeed be an artifact, possibly because the 1990s was the decade with the lowest average Tx on record, and compounded by the limited number of years available to compute reliable trends (see lack of significance of 30-year trends in the main manuscript Table 1). It is worth noting that although Tx did increase in subsequent decades, Tx was still cooler than the full-record average during 1990-2019.

To address the points brought by this comment, we have:

- Added text in the introduction to justify the need for analysis of multiple time-series
- Added a section in the methodology describing the rationale and procedures for time-series selection
- Added a few sentences in the discussion addressing the possibility that the reversal of the 30-year Tmax trend may be an artifact.
- Included these figures in the supplemental information for reference.

2. For the 2019-2050 trends, the authors do not state which time series was used for extrapolation in the text. I assume it was the 1960-2019 time period. If the authors use the 1990-2019 extrapolation

then the projection maps will look quite different. The extrapolated maps from the different time could/should go into the supplemental.

Thank you for bringing this up. We did use the trends from the three time-series to make the projections (see Fig 4) and compared them in the Results section. In the previous version of the manuscript, we opted to only show the 60-yr trends in the main text for simplicity and the other trends in the supplemental information. But we now realized that it might have conveyed the false sense that we only used the 60-yr trends for the projections when in fact we did use all. To correct this, Fig 2 and 3 now show all the trend maps, in addition to fig 4 which already did.

3. Why didn't the authors use relative humidity observations to estimate ea? Where there not enough observations?

Yes. In the ASOS data, relative humidity observations are not available for years prior to 2000. Reanalysis datasets such as GRIDMET, also have RH values, but these do not extend to years prior to 1980. Because we were interested in examining long-term trends, we decided to stick with the long-term COOP temperature records and estimate Tdew using the predictive model approach. As shown in Fig S9, this approach outperforms the assumption that Tdew equals Tmin, which is widely used in many studies.

4. There have been many studies that have examined the effects of irrigation on near-surface climate, where irrigation has shown to cool surface temperatures and decrease VPD (because of increases to humidity). How does irrigation in this region affect the results? See citations below. I would like to see a little discussion on this added to the paper.

This is a good point. Thank you for the references provided.

We are aware of the impact of irrigation on near-surface climate, and the overwhelming evidence showing that irrigation does tend to decrease VPD. We initially did not consider irrigation to be much of a factor because our analysis focuses mostly in the rainfed region of the Midwest in which irrigated areas account for less than 1% of cropland. Only the western-most fringe of our study area (Nebraska and South Dakota) has extensive irrigation in place. However, upon closer inspection of the VPD and PET trends, it was clear the areas that tended to have the greatest downward trend in evaporative demand were over these irrigated lands.

Following this comment, we decided to expand discussion by briefly reviewing evidence in support of the latent cooling effect as an explanation of the warming-hole, and the role that irrigation and cropland intensification has had on this dynamic.

5. The supplemental information document needs more detailed information. The figure captions do not give enough information to understand what the figure contains.

We apologize for this mistake. We uploaded a previous version of the supplemental information in which the captions were not complete. We have amended this error, by including the most recent version with the full captions.

Sacks, W. J., Cook, B. I., Buening, N., Levis, S., & Helkowski, J. H. (2009). Effects of global irrigation on the near-surface climate. *Climate Dynamics*, 33(2-3), 159-175.

Cook, B. I., Shukla, S. P., Puma, M. J., & Nazarenko, L. S. (2015). Irrigation as an historical climate forcing. *Climate Dynamics*, 44(5-6), 1715-1730.

Cook, B. I., McDermid, S. S., Puma, M. J., Williams, A. P., Seager, R., Kelley, M., ... & Aleinov, I. (2020). Divergent regional climate consequences of maintaining current irrigation rates in the 21st century. *Journal of Geophysical Research: Atmospheres*, 125(14), e2019JD031814.

Qian, Y., Huang, M., Yang, B., & Berg, L. K. (2013). A modeling study of irrigation effects on surface fluxes and land–air–cloud interactions in the Southern Great Plains. *Journal of Hydrometeorology*, 14(3), 700-721.

Reviewer #2 (Remarks to the Author):

Thank you for the opportunity to review this paper. It is a well written and well researched article that is suitable for publication in Nature Communications. The topic is of high importance as it discusses an implication of climate change as it relates to the highly productive agricultural region of the central U.S. I recommend the article be published with consideration of the following revisions.

I feel there should be a short discussion on crop coefficients, which could be included in lines 40 – 50 of the manuscript. ET rates vary by crop type and vary significantly over the course of the growing season. Would there be any impact on the outcome of the research if these are factored?

Thank you for this comment. Given that we used the SALUS crop model to calculate temperature trend effects on crop water stress and yield (in order to address comments made by reviewer #3), we think that discussion of crop coefficients would be redundant and unnecessary. Therefore, we decided not to include it in our revised manuscript.

Were there any quality control measures performed on the COOP and ASOS station data? The authors state the source of the data is the Iowa Environmental Mesonet program. Does this network offer quality control procedures? ASOS data in particular can be cumbersome to analyze as it includes sub-hourly data taken at what can be irregular timeframes. A brief discussion on quality of the dataset is warranted, in my opinion.

This is a good point. The Iowa environmental Mesonet computes daily summaries of both COOP and ASOS, which both are processed, and quality controlled. This service provides high-quality daily weather summaries of temperature and precipitation records with minimal or no missing data. Details about QC procedures can be seen in the mesonet website.

(<https://mesonet.agron.iastate.edu/info/datasets/climodat.html>)

In addition to the QC protocols performed by IEM, we conducted our own manual QC which included:

- Removing unreasonable values that would indicate sensor malfunction (e.g. temperatures < -45 or > 50, precipitation > 500 mm, or negative or very low RH values)
- Missing values (<0.1% of the record) were replaced with the previous day value, unless there were more than 4 consecutive days of missing data, in which case we excluded that year from the analysis.

As pointed out by the reviewer, the above procedures were more important for the ASOS dataset given the cumbersome nature of the raw data. Yet, ASOS data in our analysis had a very limited role, namely training the predictive GAM to estimate Tdew, which we showed had good performance when tested against observations of independent weather stations.

We have expanded the text in the methodology to provide more details about our quality control measures (LN 266-272).

There are several assumptions made in the calculations. What sources of errors might there be in the assumptions used? Perhaps error bars on the PET estimations might be warranted. Rs and U were fixed and do not change in the future – is this a safe assumption to make?

Thank you for this comment. As noted, treating Rs and U as fixed in our projection is an important assumption that we make in our analysis. To our knowledge, however, there is quite a lot of uncertainty on how climate change will affect cloud cover and windspeed globally, let alone in the Midwest. Thus, climate change assessments in agriculture using modeling routinely assume that radiation will remain unchanged, and that only temperature (and thus also VPD) and precipitation will be affected.

Though a link between increased cloud cover and cooler Tmax has been found for North American climates, previous research has pointed out that is unclear whether cloud cover is either the major indicator of the summer mean daily maximum temperature changes (the effect) or the important local factor influencing the changes (the cause) (Tang and Leng, 2013). Nevertheless, if we were to assume that cloudiness reduces radiation by, say, 10 or 20% in the next 30 years across the board, then our PET projections would decrease by 23 and 46 mm on average (see figure below).

However, this is probably a gross simplification, because increased cloudiness most likely reduces the number of days with clear skies, rather than making all days darker. Not to mention the negative consequences that this would have for crop yields. Because all of this is outside of the scope of our analysis, we do not incorporate this discussion into the paper other than pointing out the following:

“Other uncertainties stem from possible decreases in Rs resulting from increasing cloud cover and aerosols which were not considered in our analysis but would likely decrease PET even further.” (LN 235-236)

That being said, we appreciate the reviewer’s point about the need to characterize the uncertainties of our projections. Thus, we opted to estimate the propagation of uncertainty in the projected changes in evaporative demand (i.e. VPD) given the standard errors of the estimated temperature trends. By doing this we see that the propagated uncertainty for the 30-year time-series projection (11-32%) is about 2.5 fold that of its 60-year counterpart (5.9-13%), and about 5 fold that of the full-record projection (3.2-

6.3%). This is shown on the figure below, which is now also included in the supplementary information. We also added these results in the main text (LN 129-135)

Finally, it might be good to add a few more items to the discussion related to climate change implications for agriculture – namely potential of erosion from heavy rain events, timing of heat events in relation to corn pollination, lengthened growing season, etc.

We appreciate the spirit of this comment, and very much agree that climate change remains a challenge for agriculture in the Midwest, regardless of the positive benefits it might have for crop water availability. And by no means we aim to undermine the sense of urgency for addressing its root causes.

We could indeed include a sentence on this to underline that these findings do not solve the major threats of climate change related to soil erosion, extreme events, etc. However, we believe that this would not add much to the discussion given that this was not the topic of the study and several papers have specifically addressed these major challenges.

However, to address this comment, we opted to add a sentence related to the fact that even though we predict decreases in the mean PET, this does not necessarily mean less risk of drought or changes in the timing of rainfall that could affect sensitive periods such as pollination or kernel set. (LN 211-214)

Good luck with the publication process,
Martha Shulski

Reviewer #3 (Remarks to the Author):

In this MS, the authors investigate past long-term trends in temperature and future changes in potential evapotranspiration in the agriculture region of Midwest US. The paper presents a topic which is of great interest within the agricultural communities given the potentially significant changes in crop growth that could occur as a result of ongoing climate warming.

The paper is very well written. However, this analysis should be extended to demonstrate the added value of the trend analysis beyond just using the methods for projecting changes in summer PET. I didn't find the results too intriguing since they essentially showed the impacts of contrasting trends in T_x and T_{dew} and how these trends would affect the growing season PET. I would have expected the authors to have carried out additional analysis of changes in the water deficit combined with trends in crop yield and come up with more scientific statements about how the crop growth could be affected. As currently presented, this work provides no useful information that can benefit farmers in their attempt to cope with the projected changes due to climate change.

I would like to see at least some additional analysis of crop yield trends to also show how the changes in temperature and PET would influence the crop growth for understanding of past interactions between yield and climate trends but also the future changes. If the authors have more results in this vein, they should upgrade their study accordingly.

Thank you for your comments. Indeed, we agree that adding further analysis about the potential impacts of climate trends in the Midwest on crop yields would add value and provide more context to our PET projections.

Previous work has in fact already retrospectively examined the impact of the 'warming-hole' climate trends on crop yields in the Midwest, showing that these trends boosted crop yields by 5 to 10% (Partridge et al. 2019). Here we build on these findings and focus specifically on the yield impacts given our projections derived from the computed trends.

To address these comments, we opted for a dynamic modelling approach to simulate simulating corn growth under historical and 2050 climate scenarios for the three time-series trends. We use the SALUS process-based crop model which has been shown to be capable of capturing soil hydrology and evapotranspiration dynamics across many crops and soils. This model was checked against observed yields as reported by the National Agricultural Statistics Service (NASS) as well as compared with the Δ PET computed offline by Penman-Monteith, both of which showed good agreement. More details about the model and the implementation procedures were included in the methodology (LN 361-378).

The result of the simulations is show in the following figures which are now Fig 5 and 6 in the revised manuscript.

Based on this new data, we conclude that these projected changes in PET and precipitation could largely reduce corn water deficits and have a neutral to slightly beneficial effect on corn yields. Even in the cases when temperature trends do lead to increases in PET in our simulations, impacts on corn yields are expected to remain small (<6%).

Additionally, though the simulated responses in corn water stress and yields reflects changes in both JJA PET and rainfall, changes in water stress were more strongly correlated with changes in total rainfall than corn yield; whereas yield showed stronger correlation to changes in PET. All of these results are presented at length (LN 137-164) and discussed (LN 207-210, 237-241) in the revised manuscript.